# The distinct role of human PIT in attention control

**Siyuan Huang[1,2], Lan Wang[1], Sheng He[1,2,3]***

[1]State Key Laboratory of Cognitive Science and Mental Health, Institute of Biophysics, Chinese Academy of Sciences, Beijing, China; [2]University of Chinese Academy of Sciences, Beijing, China; [3]Institute of Artificial Intelligence, Hefei Comprehensive National Science Center, Hefei, China

## eLife Assessment

This **important** study reports that the human posterior inferotemporal cortex (hPIT) functions as an attentional priority map, integrating both top-down and bottom-up attentional signals rather than serving solely as an object-processing region. The experiments and analyses are well conducted and provide **compelling** evidence that hPIT bridges dorsal and ventral attention networks and is robustly modulated by attention across diverse visual tasks. The study will be relevant for researchers investigating visual attention, high-level visual cortex, and the neural mechanisms that integrate endogenous and exogenous attentional control.

***For correspondence:**
hes@ibp.ac.cn

**Competing interest:** The authors declare that no competing interests exist.

## Abstract

Attentional distribution depends on both endogenous and exogenous processes, but how they interact in attentional allocation remains unclear. The attentional priority map, jointly determined by stimulus saliency and task relevance, provides a framework for investigating their interplay. We propose that the human posterior inferotemporal cortex (hPIT), located near object-processing cortical areas, serves as an attentional priority map. Using fMRI with behavioral tasks, we show that hPIT shows stronger attentional modulation than classical attentional regions across motion, color, and shape tasks. hPIT shows lateralized attentional enhancement even in the absence of visual input, with further elevated modulation in the presence of stimuli, indicating its integrated role in priority control. Furthermore, its modulation is invariant to stimulus category but sensitive to attentional demands, and the region is functionally connected to both dorsal and ventral attentional networks. These findings highlight the hPIT as an integrator in attentional control and provide critical insights into the brain's strategy for optimizing responses to the environment.

## Introduction

Attention serves as a pivotal mechanism within the brain, facilitating the allocation of its finite resource toward the most pertinent stimuli in our environment (*Desimone and Duncan, 1995*). Attention can be systematically categorized according to its origin: bottom-up attention (exogenous), triggered by highly salient stimuli, and top-down attention (endogenous), initiated through voluntary selection (*Connor et al., 2004*; *Buschman and Miller, 2007*; *Katsuki and Constantinidis, 2014*). The neural networks for bottom-up and top-down attention have been extensively investigated, with ventral and dorsal attentional networks (VAN and DAN) specialized to support exogenous and endogenous attention, respectively (*Corbetta and Shulman, 2002*; *Corbetta et al., 2008*; *Vossel et al., 2014*). How these two networks flexibly interact to achieve integrated attentional selection, under situations that both salient stimulus and intentional goal are present, remains inadequately understood (*Noudoost et al., 2010*).

Critical to the integration of exogenous and endogenous attention is the concept of 'priority map', defined as a map that reflects both the low-level salience and top-down influences (*Treue, 2003*; *Bisley and Goldberg, 2010*; *Bisley and Mirpour, 2019*). An important question arises concerning the location of the priority map in the brain. For a brain region to qualify as the neural substrate supporting a priority map, it must satisfy the following criteria (*Ptak, 2012*; *Stemmann and Freiwald, 2019*): (1) having spatially restricted receptive fields, (2) exhibiting robust attentional modulation to predict attentional focus regardless of stimulus attributes but sensitive to attentional load, (3) displaying little feature specificity and high visual responsivity, and (4) exhibiting integration of bottom-up and top-down inputs. While the saliency map – where objects compete for greater representation and attentional allocation (*Fecteau and Munoz, 2006*) across all feature dimensions – plays an important role in the mechanism of bottom-up attention, the priority map (jointly determined by saliency and task guidance) is crucial for the cooperation of bottom-up and top-down attention, making it a key aspect to understand for the implementation of attention.

Previous research has indicated that the prefrontal cortex (PFC) and posterior parietal cortex are involved in both top-down and bottom-up attention, rendering them potential candidates for supporting a priority map guiding attentional selection (*Herrington and Assad, 2009*; *Arcizet et al., 2011*; *Katsuki and Constantinidis, 2012*; *Ninomiya et al., 2012*; *Katsuki and Constantinidis, 2014*). In addition, the superior colliculus (SC), especially its intermediate layers, receives signals from the frontal eye fields (FEF) and lateral intraparietal area (LIP)/intraparietal sulcus (IPS) (*Rushworth et al., 2006*; *Schall et al., 2011*; *Benarroch, 2023*), could also support a priority representation (*White et al., 2017*). While visual areas, driven by stimulus features, were considered unlikely to exhibit priority representation (*Bisley and Mirpour, 2019*), a recent study in monkeys revealed that the posterior inferotemporal cortex (PITd) could encode the locus of attention independent of features (*Stemmann and Freiwald, 2019*). The structural connectivity between PITd and parieto-frontal attentional areas (specifically LIP and FEF) further supports the idea that PITd is part of attentional networks (*Sani et al., 2019*). In the context of human brains, recent results have identified an area in the human posterior inferotemporal cortex (hPIT) as a retinotopic and functional homologue of the macaque PITd, which was proposed to be a putative node for the human endogenous attentional control network (*Sani et al., 2021*). However, since only endogenous attention was investigated in the previous experiments on monkey PITd, as well as hPIT, it remains unclear whether hPIT is important for the integration of endogenous and exogenous attention.

The current study investigated if attention can modulate activation in hPIT independent of stimulus properties and cognitive demands. Instead of relying solely on the attentive motion task (*Sani et al., 2021*), hPIT was localized and validated using three distinct spatial attentional tasks. Furthermore, we aim to discern the role of hPIT in both top-down and bottom-up attention by manipulating the presence or absence of visual stimuli. Distinct from nodes of classical endogenous attentional network, hPIT was more strongly modulated by attention in the presence of visual input. Furthermore, attentional load, rather than object category, significantly impacts the modulation in hPIT. Additionally, hPIT showed strong functional connection with nodes in both VAN and DAN. Our results strongly suggest that hPIT, different from its adjacent object-processing areas and other parietal and frontal nodes of attentional network, functions as an attentional priority map.

## Materials and methods

### Participants

Fifteen health volunteers (8 males and 7 females, age ranged from 22 to 28 years of age) with normal or corrected to normal vision participated in this study. None of the participants reported a history of neurological or psychiatric symptoms. Each participant completed three separate experimental sessions, conducted on 3 different days. Written informed consent was obtained from all participants, and the study protocol was approved by the Institutional Review Board of the Institute of Biophysics, Chinese Academy of Sciences (#2020-IRBH-001).

### Stimuli and procedures

Visual stimuli for the fMRI experiment were programmed using MATLAB (MathWorks, Natick, MA, USA) with Psychtoolbox (http://psychtoolbox.org/) and displayed via an MRI-compatible projector

(1024×768@60 Hz) onto a screen positioned at the rear of the MRI scanner. Participants viewed the stimuli through a mirror attached to the head coil.

Because of the complexity of the task, each participant was trained on 2 or 3 separate days, before the fMRI sessions. This training was designed to familiarize participants with the tasks to relieve some possible tension during scanning and minimize the effects of learning.

## Stimuli and procedures for Experiment 1

In Experiment 1 (day 1), participants completed three distinct spatial attentional tasks across three separate experimental blocks (Experiments 1a, 1b, and 1c). Throughout each task, participants were instructed to maintain fixation on a central point.

### Experiment 1a (Figure 1A)

Participants focused on a unilateral moving dot display and were required to discriminate the direction of the coherent motion. As shown in *Figure 1A*, each trial began with an initial cue indicating the target side, represented by a bar (0.6° visual angle) attached to the fixation point. Random dot stimuli were then presented within a circular aperture (radius: 3° visual angle) centered 9° from the fixation point. The dot display consisted of dots (size: 0.15° visual angle; density: 5 dots per degree of visual angle) in motion for 2.5 s. During this sequence, a 0.5 s interval of coherent motion (coherence: 50%, velocity: 6°/s) occurred randomly between 0.5 and 1.55 s after motion onset, while the remaining duration involved random dot movement. Following the motion sequence, the circular aperture disappeared, and an arrow appeared at the center of the screen for 1.5 s. Participants were required to press a key to indicate whether the direction of coherent motion matched the direction indicated by the arrow. Each block consisted of three trials with the same attended side, followed by a 6 s rest period.

### Experiment 1b (Figure 1B)

Participants were instructed to attend to the proportion of red and green dots presented within a unilateral circular aperture. The red and green dots were adjusted individually by each participant to achieve iso-luminance, ensuring perceptual equality between the colors. The positioning of the circular aperture was consistent with Experiment 1a, though the dot size was increased to 0.3° of visual angle in this task. At the start of each trial, within the first second, dots were displayed in random proportions of red (20%, 40%, 60%, 80%) with the remainder being green (pattern 1). Following this, the position of the dots remained static during the 2nd second, but their color could potentially change (pattern 2). The proportion of red dots remained to be one of the pre-set ratios. In the final, third second of each trial, the bilateral dots disappeared. Participants were then required to press a key to indicate whether the color ratio of the dots had changed from pattern 1 to pattern 2. Each block in this experiment consisted of four trials, with participants attending to the same side throughout, followed by a 6 s rest period.

### Experiment 1c (Figure 1C)

In this experiment, participants were instructed to attend to a unilateral geometrical pattern display, focusing on the proportion of different shapes. Each trial began with the first pattern presented for 2 s, followed by a gradual transition in which the second pattern emerged over the next 2 s and then remained visible. At the end of this sequence, participants were required to press a key indicating whether the shape proportion between the two patterns had changed. Each block consisted of two trials with attention directed to the same side, followed by a 6 s rest period.

There were 16 blocks in 1 run, 8 attending left and 8 attending right for Experiments 1a, 1b, and 1c. On the first day of scanning, participants completed 6 runs of Experiment 1a, 4 runs of Experiment 1b, and 4 runs of Experiment 1c. The average accuracy was 95% for Experiment 1a, 90% for Experiment 1b, and 90% for Experiment 1c.

## Stimuli and procedures for experiment 2

In Experiment 2 (day 2), participants first underwent a 488 s resting-state functional scan, during which they passively viewed a gray screen. For the subsequent attentional tasks (event-related design),

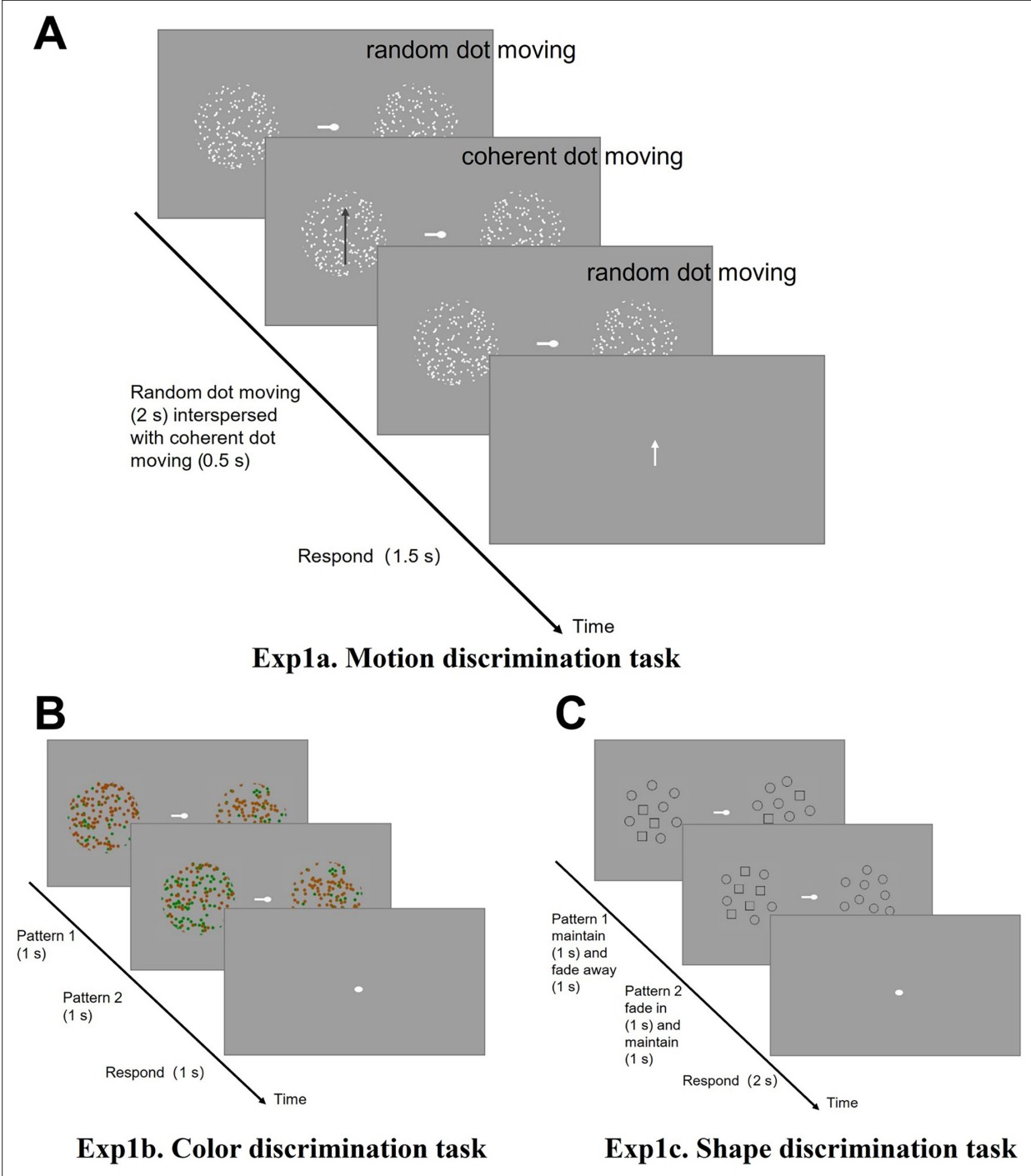

**Figure 1.** Schematic representation of the experimental paradigm for Experiment 1. (**A**) The task was to report whether the direction of coherent motion on the attended side matched that of the white arrow. Note: The black arrow, representing one potential direction for the coherent dot movement, is used for illustrative purposes and was not actually presented during the experiment. (**B**) Pattern 1 and pattern 2, consisting of iso-luminant red and green dots, were presented sequentially. The task was to compare the color ratios of these patterns on the attended side and respond accordingly if any changes in the color ratio were detected. (**C**) Pattern 1 and pattern 2, consisting of equal number of small shapes (circles and squares), were presented sequentially. The task was to compare the shape ratios of these patterns on the attended side and respond accordingly if any changes in the shape ratio were detected.

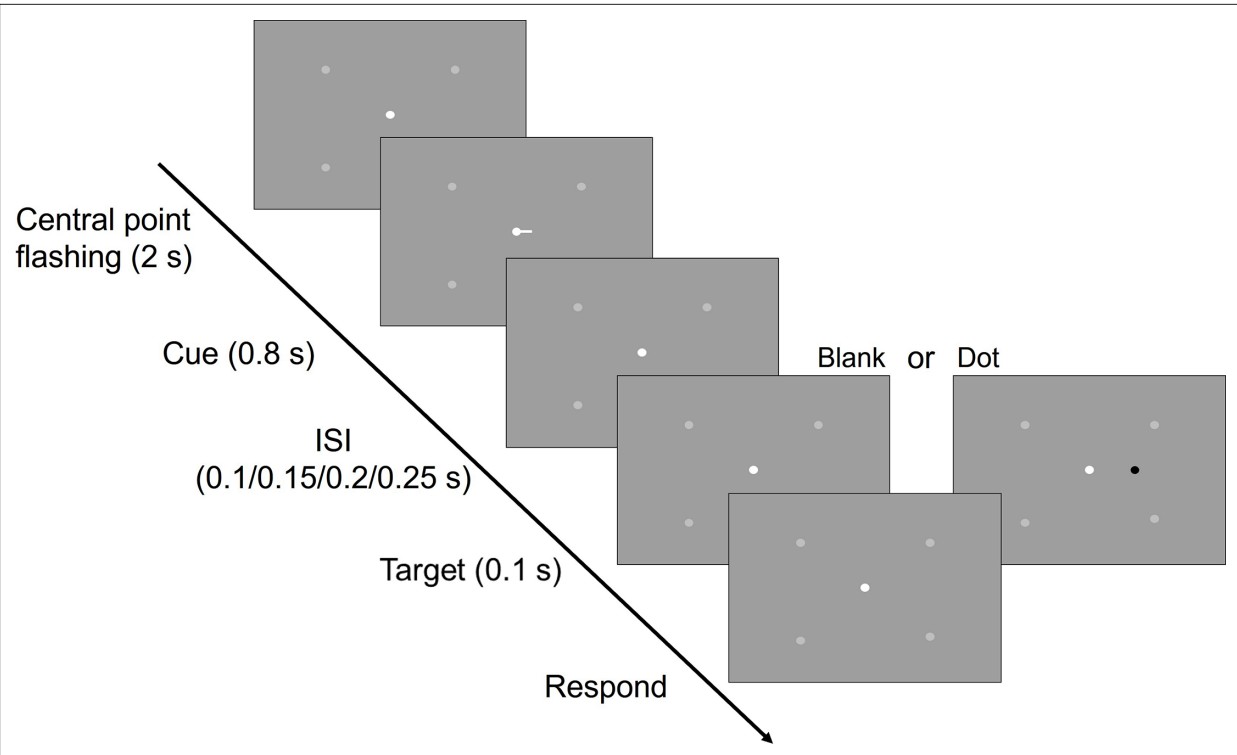

**Figure 2.** Schematic depiction of the experimental design for Experiment 2. Following the flashing of the central point, a cue is presented. In the dot condition, a dot appears on the attended side, prompting participants to report its relative position. In the blank condition, participants are instructed not to press any keys. (Light gray dots: serve as indicators, showing participants the potential target locations during the experiment.)

participants were instructed to maintain fixation on a central dot while directing their attention to one side.

The procedure (*Figure 2*) began with the central dot (10 pixels in diameter) flashing three times over 2 s, serving as an alert for the upcoming cue and target. Following this, a cue was presented for 0.8 s. After a variable delay, randomly selected from intervals of 0.1, 0.15, 0.2, or 0.25 s, a target dot (8 pixels in diameter, RGB: [0.7,0.7,0.7]) either appeared on the cued side (dot condition) or did not appear (blank condition). In the dot condition, participants were required to report the location of the target by pressing a key. In the blank condition, participants were instructed to refrain from any response.

Each run consisted of 16 trials, with individual trial durations of 14, 16, or 18 s. On this second day of scanning, participants completed 1 run of the resting-state scan, followed by 8 task runs.

## Stimuli and procedures for Experiment 3

In Experiment 3 (day 3), participants were asked to attend unilateral images presented within a circular aperture, consistent with the method used in Experiment 1. The images were drawn from three distinct categories:

Faces: Asian faces with an equal gender distribution (50% female).
Scenes: Equally divided between indoor and outdoor settings (50% outdoor).
Scrambles: Phase-scrambled images superimposed with a grating, with 50% of the gratings spanning the first and third quadrants.

To ensure visual consistency across images, they were standardized for luminance histograms and spatial frequency using the SHINE toolbox (as referenced from *Willenbockel et al., 2010*).

At the onset of each run, participants were instructed to pay extra attention to a designated category throughout that run. The procedure (*Figure 3*) began with the center dot flashing three times over 2 s, signaling the forthcoming presentation of the cue and target. Following this, both the cue and bilateral images from the same category were presented simultaneously. These images

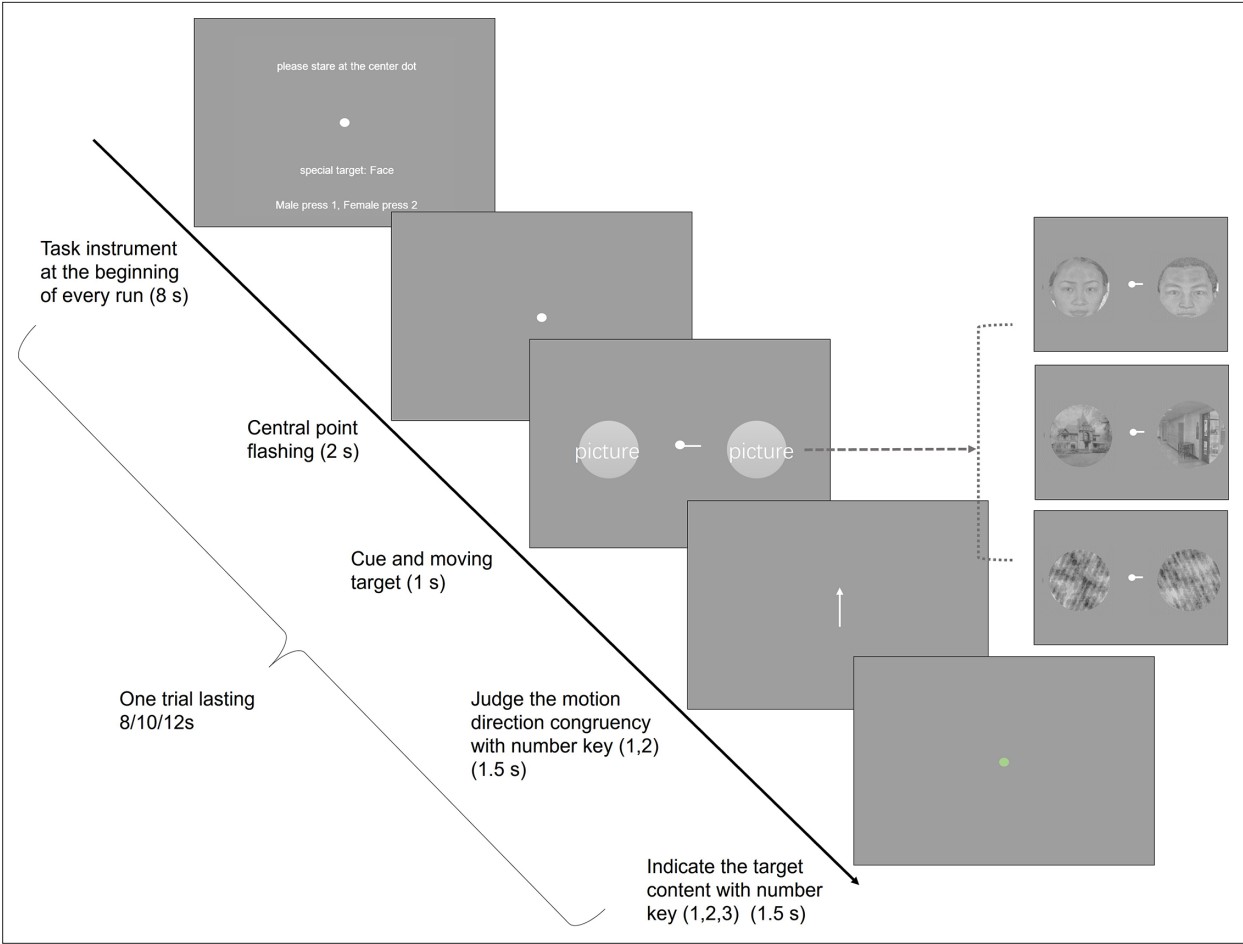

**Figure 3.** Schematic overview of the experimental paradigm for Experiment 3. At the start of the 'face run', participants were instructed to pay heightened attention to the face images throughout that specific run. Following the central point's flashing sequence, a cue and bilateral images were presented. Participants first judged if the movement direction of the attended image aligned with the central arrow. Upon the central point changing to green, participants then responded based on the content of the attended image.

briefly shifted in one direction before returning to their original position (0.25 s+0.25 s). During the next 1.5 s, participants first pressed a key to indicate whether the direction of the images' motion matched that of the arrow at the center. A green dot then appeared at the center, reminding participants to categorize the image in their attended location. If the image matched the category to which they had been instructed to pay attention, participants were required to identify specific content about the image (e.g. determining whether a face was female or male) and responded by pressing either key '1' or '2'. If the image did not belong to the designated category, participants simply pressed '3'.

Each run consisted of 32 trials: 50% of the trials featured images from the emphasized category, and the remaining 50% were evenly distributed between the other two categories (25% each). Each trial lasted either 8, 10, or 12 s. Across Experiment 3, there were a total of 9 runs, with each category being the focus of 3 runs. The accuracy of extra judgments is 82.9% for face, 81.8% for scene, and 84.3% for scramble ($F_{(1.373, 19.23)}=1.096$, $p=0.3308$).

## MRI data acquisition

MRI scanning was conducted using a 3T Siemens Prisma scanner at the Beijing MRI Center for Brain Research (BMCBR), utilizing a standard 20-channel head coil. High-resolution T1-weighted anatomical images were acquired at the start of each session (TR = 3000 ms; TE = 3.02 ms; 176 slices; slice thickness = 1 mm; no inter-slice gap; field of view = 256 mm; flip angle = 8°; image matrix: 256×256).

Functional data were collected with gradient-echo EPI sequences (TR = 2000 ms; TE = 30.0 ms; 52 slices; slice thickness = 2 mm; no inter-slice gap; voxel resolution 2.0×2.0×2.0 mm³, field of view = 192 mm; flip angle = 80°; image matrix: 96×96).

## MRI data analysis

fMRI data were preprocessed and analyzed using FreeSurfer and AFNI and custom Python code (primarily the mripy package, https://github.com/herrlich10/mripy; *herrlich10, 2025*). The preprocessing steps included de-spiking, slice timing correction, EPI distortion correction (PE blip-up), rigid body motion correction, spatial smoothing (4 mm FWHM Gaussian kernel, for the task runs), and per run scaling (as percent signal change).

For task runs, general linear models were used to estimate BOLD signal change from baseline for each stimulus condition. For each individual, bilateral fusiform face area (FFA), occipital face area (OFA), and parahippocampal place area (PPA) were defined based on the functional contrast between faces and scenes from Experiment 3. Bilateral V1 was defined using functional contrast (attend contra moving dot – baseline) (p<0.001, uncorrected), and IPS, FEF, and MT were defined using the same contrast (p<0.01, uncorrected) and referring to the cerebral atlas (*Glasser et al., 2016*). Due to the lack of standardized anatomical definitions, bilateral ventral frontal cortex (VFC) and temporal parietal junction (TPJ) were identified using the same functional contrast (p<0.01, uncorrected) (*Vossel et al., 2014*). Bilateral hPIT was localized using data from all three tasks in Experiment 1, by intersecting activation maps on the inferotemporal surface, while avoiding the locations of FFA and PPA.

The resting-state data was preprocessed similarly to the functional data, with additional steps for removing white matter and cerebrospinal fluid signals. Subsequently, confound regression, spatial smoothing (4 mm), and bandpass filtering (0.01–0.1 Hz) were performed using AFNI's 3dTproject.

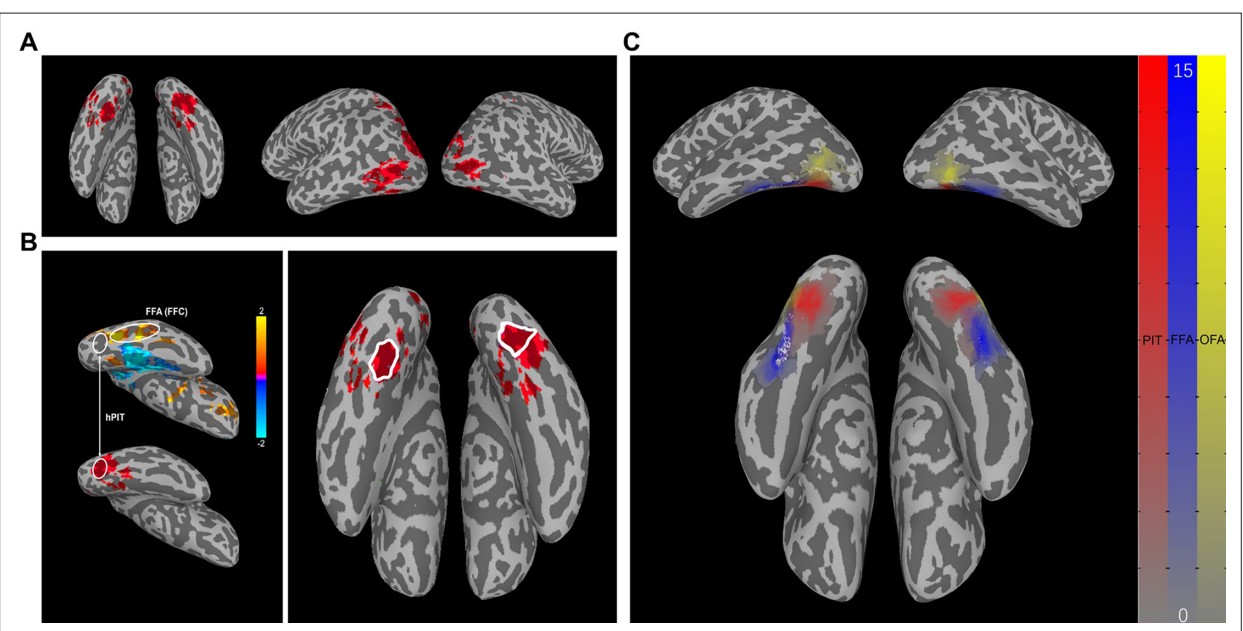

**Figure 4.** Functional localization of human posterior inferotemporal cortex (hPIT). (**A**) The intersection of three maps of three block tasks on one typical participant (S02). The cortical areas which are attached in red have shown significant activation in all three attentional tasks. (**B**) The exact position of hPIT. Left: Positions of fusiform face area (FFA) and hPIT in statistical parametric maps of the contrast (attend face – attend scene) (p=0.05) (top) and hPIT in intersection map on the right hemisphere of S02. Right: The location of hPIT on both hemispheres of S02 is circled by a white line. The cortical areas which are attached in red have shown significant activation in all three attentional tasks. (**C**) The positions of hPIT, occipital face area (OFA), and FFA of 15 participants overlapped on the surface of MNI152_2009c, with larger numerical values (manifested as deeper colors) indicating a higher degree of overlap among participants. The color scale ranging from gray to red delineates the spatial distribution of the hPIT, while the scale from gray to blue represents the FFA, and gray to yellow signifies the OFA.

## Results

### Localization and validation of hPIT from task-invariant activation to spatial attention

To identify brain area(s) that encode the location of attentional invariant to stimulus type and cognitive demand (*Stemmann and Freiwald, 2019*), we employed three different spatial attentional tasks in Experiment 1. In all three tasks, participants were instructed to fixate on a center dot and pay attention to the circular aperture on one side, as directed by the cue. First task was about motion discrimination, and the second and third tasks required the discrimination of color proportion and shape proportion.

To isolate cortical areas modulated by endogenous spatial attention, contrasts between the conditions (attend contralateral – attend ipsilateral) were calculated for each of the three task blocks. Significant attention-modulated voxel maps were generated for individual participant (in most cases $p<0.05$, cluster size >20, but for activated voxel number >10,000, using $p<0.01$). Then, we took the intersection of these three maps and projected it to the cortex in every participant (example shown in *Figure 4A*), as our goal was to locate hPIT in inferotemporal cortex of human brain which should exhibit the properties of priority map, specifically in this context, insensitive to stimulus and cognition dimension. On the intersection of three maps, we found bilateral brain areas significantly modulated by spatial attention in intraparietal sulcus (excluding the right hemisphere of S13) and inferotemporal cortex in all participants, and in most participants also the middle temporal cortex (excluding S07). Only three participants (S03, S04, S11) showed intersection area in the PFC.

As illustrated in *Figure 4B*, the hPIT was manually identified bilaterally on the intersection maps within the posterior and ventral part of mid-fusiform sulcus as a contiguous cluster of voxels that are non-overlapping with FFA nor PPA. In all participants, the hPIT is ventromedial to the OFA and posterior to the FFA. The locations of hPIT, OFA, and FFA identified in our study are shown in *Figure 4C* on the surface of standard brain MNI152_2009c. The MNI coordinates delineating the center of mass for the ROIs are specified as follows: hPIT (−34, −72, −14), (33, −73, −13); FFA (−43, −55, −19), (41, −54, −17); OFA (−46, −79, −3), (46, −74, −8). The average cluster size of the hPIT is 210 voxels in the left hemisphere and 191 voxels in the right hemisphere, with 2×2×2 mm³ voxels. Detailed location and cluster size of hPIT in every participant's cortical surface are reported in *Supplementary file 1*.

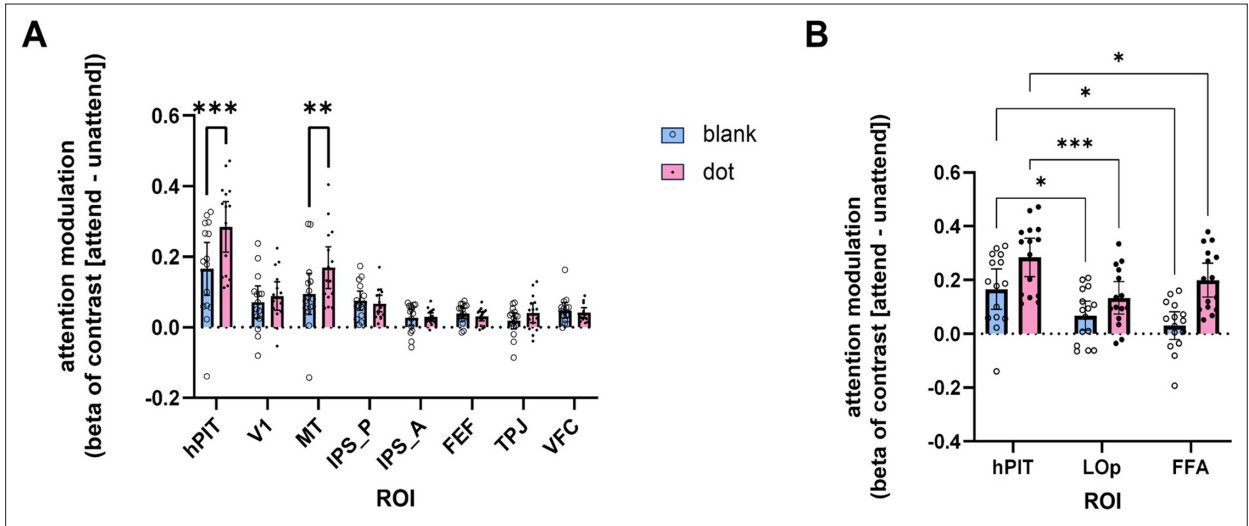

**Figure 5.** Attentional modulation in different brain regions. (**A**) The modulation pattern of V1, human posterior inferotemporal cortex (hPIT), medial temporal visual area (MT), intraparietal sulcus (IPS), frontal eye fields (FEF), temporal parietal junction (TPJ), and ventral frontal cortex (VFC) in condition blank (blue bar) and condition dot (pink bar), using beta of contrast: (attended – unattended [attend contralateral – attend ipsilateral]). The modulation difference of attention between condition blank and condition dot reached a significant level in posterior inferotemporal cortex (PITd) and MT. (**B**) The modulation pattern of hPIT, lateral-occipital cortex (LOp), and fusiform face area (FFA) in condition blank (blue bar) and condition dot (pink bar). Error bars indicate 95% confidence interval (n=15). Statistical significance was assessed using a two-tailed paired t-test. *** $p<0.001$; ** $p<0.01$; * $p<0.05$.

The online version of this article includes the following figure supplement(s) for figure 5:

**Figure supplement 1.** The top-down attentional modulation (attend – baseline) in bilateral frontal and parietal regions during the blank condition.

Enhanced activation to attended than unattended moving dots in Experiment 1, constrained by cerebral atlas, allowed us to also locate cortical areas that are critical nodes of attentional network (*Corbetta and Shulman, 2002*; *Vossel et al., 2014*) bilaterally, including medial temporal visual area (MT), IPS (classified into IPS_P, the posterior part, and IPS_A, the anterior part), FEF, TPJ, and VFC (see Materials and methods). Additionally, V1 was included as an ROI, so that we can see the consequences of attentional modulation in the primary visual cortex. Furthermore, the posterior part of the lateral-occipital cortex (LOp) and FFA were localized using cortical parcellation and functional contrast as controls, since they were adjacent to hPIT physically.

## Attentional modulation of hPIT with and without bottom-up input

To investigate the modulation in hPIT and other brain areas applied by top-down attention (without stimulus) and top-down combined with bottom-up attention (with stimulus), we manipulated the presence or absence of visual stimuli in Experiment 2. Using event-related design, participants were required to direct their attention to the left or right visual field based on central cues. A dot was then presented on the attended side on half of the trials (50% probability), requiring participants to report, by pressing keys, its location relative to two reference points that were constantly presented (*Figure 2*).

ROI-based analysis was performed to BOLD activation signals from hPIT and other ROIs identified in Experiment 1. *Figure 5A* shows the averaged beta of contrast (attended – unattended [attend contralateral – attend ipsilateral]) from bilateral ROIs for each participant, illustrating activation patterns of our ROIs under both blank and dot conditions. A two-way ANOVA (n=15), considering blank/dot and ROI as factors, revealed significant main effects for both: ROI ($F_{(2.246, 31.44)}$=19.20, $p<0.0001$) and condition blank/dot ($F_{(1.000, 14.00)}$=20.41, $p=0.0005$). A significant interaction between these two factors was also observed ($F_{(4.061, 56.86)}$=10.98, $p<0.0001$).

The condition wherein participants were required to attend to one side in the absence of any stimulus except the constant background served to elucidate the impact of top-down attentional modulation. As illustrated by the blue bars (*Figure 5A*), top-down modulation was apparent in multiple ROIs (V1, hPIT, MT, IPS_P, IPS_A, FEF, VFC), but it was strongest in hPIT. To establish quantitative metrics, a post hoc multiple comparison test (Dunnett, one-sided) was utilized to compare the strength of attentional modulation between hPIT and other ROIs. The attentional modulation in hPIT was significantly stronger than all the other ROIs: V1 ($q_{(14)}$=2.648, adjusted $p=0.0451$), MT ($q_{(14)}$=2.856, adjusted $p=0.031$), IPS_P ($q_{(14)}$=2.680, adjusted $p=0.0426$), IPS_A ($q_{(14)}$=3.883, adjusted $p=0.005$), FEF ($q_{(14)}$=3.823, adjusted $p=0.005$), TPJ ($q_{(14)}$=4.261, adjusted $p=0.002$), VFC ($q_{(14)}$=3.161, adjusted $p=0.018$). Critically, since higher-order regions may exhibit large receptive fields potentially reducing the degree of contralateral bias, we specifically evaluated whether these frontal and parietal regions (FEF, VFC, IPS_A, IPS_P) demonstrated engagement during top-down attention. One-sample t-tests confirmed significantly greater activation during the attend-blank condition compared to baseline (all $p<0.05$; see *Figure 5—figure supplement 1*), providing direct evidence for their involvement in top-down attentional deployment.

With the introduction of a small dot as a target stimulus, we were able to examine the combined influence of top-down and bottom-up attention. As indicated by the pink bars, hPIT again exhibited the strongest attentional modulation effect; however, here the modulation effect reflects the combined effect of top-down and bottom-up attention. Post hoc comparison (Dunnett, one-sided) revealed again that hPIT had significantly higher combined attentional modulation effect than all the other ROIs: V1 ($q_{(14)}$=5.717, adjusted $p<0.001$), MT ($q_{(14)}$=5.443, adjusted $p<0.001$), IPS_P ($q_{(14)}$=6.742, adjusted $p<0.001$), IPS_A ($q_{(14)}$=7.804, adjusted $p<0.001$), FEF ($q_{(14)}$=7.939, adjusted $p<0.001$), TPJ ($q_{(14)}$=7.944, adjusted $p<0.001$), and VFC ($q_{(14)}$=7.554, adjusted $p<0.001$). Further, among all the ROIs, hPIT and MT showed significantly stronger elevation in attentional modulation in the dot than the blank condition (Bonferroni, hPIT: $t_{(14)}$=5.321, adjusted $p<0.001$; MT: $t_{(14)}$=4.326, adjusted $p=0.003$). Comparatively, the combined attentional effect was stronger in hPIT than in MT (paired t-test, one-tailed, $t_{(14)}$=1.969, $p=0.035$). Specific values representing attentional modulation for all ROIs (V1, MT, IPS, FEF, TPJ, VFC, FFA, LOp) under both blank and dot conditions are provided in *Supplementary file 2*.

As a supplement, comparison of attentional modulation across ROIs adjacent to hPIT was conducted (*Figure 5B*). Results of two-way ANOVA showed significant main effect of condition blank/dot (df =

1, F(1.000, 14.00)=30.23, p<0.0001) and ROIs (df = 2, F(1.921, 26.90)=9.788, p=0.0007), as well as the interaction (df = 2, F(1.311, 18.36)=5.691, p=0.0210). Multiple comparisons (Tukey's) demonstrated that the modulation in hPIT was significantly stronger than LOp and FFA in both conditions (Blank: q(14)=3.822, p=0.043 for LOp, q(14)=4.581, p=0.015 for FFA; Dot: q(14)=7.054, p=0.001 for LOp, q(14)=4.137, p=0.028 for FFA). These findings highlight hPIT's distinctive pattern of attentional engagement compared to the other two areas.

Complementary analysis revealed hemispheric differences in hPIT attentional modulation. A two-way ANOVA (n=15, hemisphere × condition) was performed, results showed significant main effect of hPIT hemisphere (df = 1, F(1.000, 14.00)=7.816, p=0.0143), significant main effect of condition (df = 1, F(1.000, 14.00)=28.31, p=0.0001) and nonsignificant interaction (df = 1, F(1.000, 14.00)=0.9477, p=0.3468).

Generally, while most ROIs showed significant modulation by attention in both the blank and dot conditions (see *Supplementary file 3*), hPIT is unique in that it showed the strongest attentional modulation in each condition, as well as showing the largest elevation effect from blank to dot condition, suggesting that hPIT is deeply engaged in both bottom-up attention and top-down attention.

## Image category-invariant response but load-sensitive attentional effect in hPIT

To effectively guide spatial attention, hPIT should be broadly responsive to different stimulus features (i.e. lack of feature selectivity) but show high sensitivity to locations with salient and task-relevant features. In addition, such spatial sensitivity should be modulated by attentional load. In order to explore stimulus feature and attentional load sensitivity of hPIT, in Experiment 3, images of three different categories (face, scene, scramble) were presented, participants were required to pay attention to the image on the left or right side according to the cue and report its moving direction. To modulate the attentional load, in each run, participants were instructed to make more demanding judgments (higher load) about the content of one pre-specified category of the attended images. For the pre-specified category, the additional responses were female or male for faces, indoor or outdoor for scenes, overlay grating tilted clockwise or counterclockwise for the scrambled images (*Figure 3*).

*Figure 6A* shows the response of hPIT to different categories of images in different conditions, indicated by the beta values. To test if hPIT showed any effect of category preference, spatial attention, and attentional load sensitivity, we first performed a three-way ANOVA (n=15, spatial attention × stimulus category × task load). Results showed that the main effect of attention was significant (F(0.7628, 10.68)=119.8, p<0.0001), the interaction between attention and category was significant (F(2.000, 28.00)=4.068, p=0.0281), and the interaction between attention and task load was significant (F(0.6964, 9.749)=8.307, p=0.0230). Other main effects and interactions did not reach significance.

Since there is attention × category interaction, we further explored stimulus category sensitivity separately in the attend and unattend conditions. Bayesian repeated measures ANOVA was adopted to compare the responses to face, scene, and scramble stimuli under attend and unattend conditions: attend (BF$_{10}$=0.93), unattend (BF$_{10}$=0.202). The small Bayesian factors (<1) under both conditions provide support that hPIT response was insensitive to these three categories, consistent with the criteria for priority map.

In addition, as the main effect of attention is significant, to gain a better understanding of the attentional effect related to stimulus categories, we calculated the attentional modulation of hPIT for different image categories and task load (*Figure 6B*). A two-way ANOVA (n=15, stimulus category × task load) was performed. Significant main effects of category (F(1.852, 25.93)=4.069, p=0.0317) and task load (F(1.000, 14.00)=8.307, p=0.0121) were found while their interaction was nonsignificant (F(1.527, 21.37)=2.714, p=0.1004). Further investigating whether the observed attentional load effect (difference between high-load and low-load condition) differed between hemispheres, we directly compared the load modulation (averaged across stimulus categories) in left and right hPIT (*Figure 6C*). This comparison revealed no significant difference between hemispheres (two-tailed paired t-test: t(14)=0.4791, p=0.6393).

In general, unlike its adjacent object-processing areas, the response of hPIT is not sensitive to stimulus category. There is evidence that attentional modulation in hPIT is slightly stronger to faces than scenes and scrambled images, possibly reflecting the intrinsic saliency difference between these

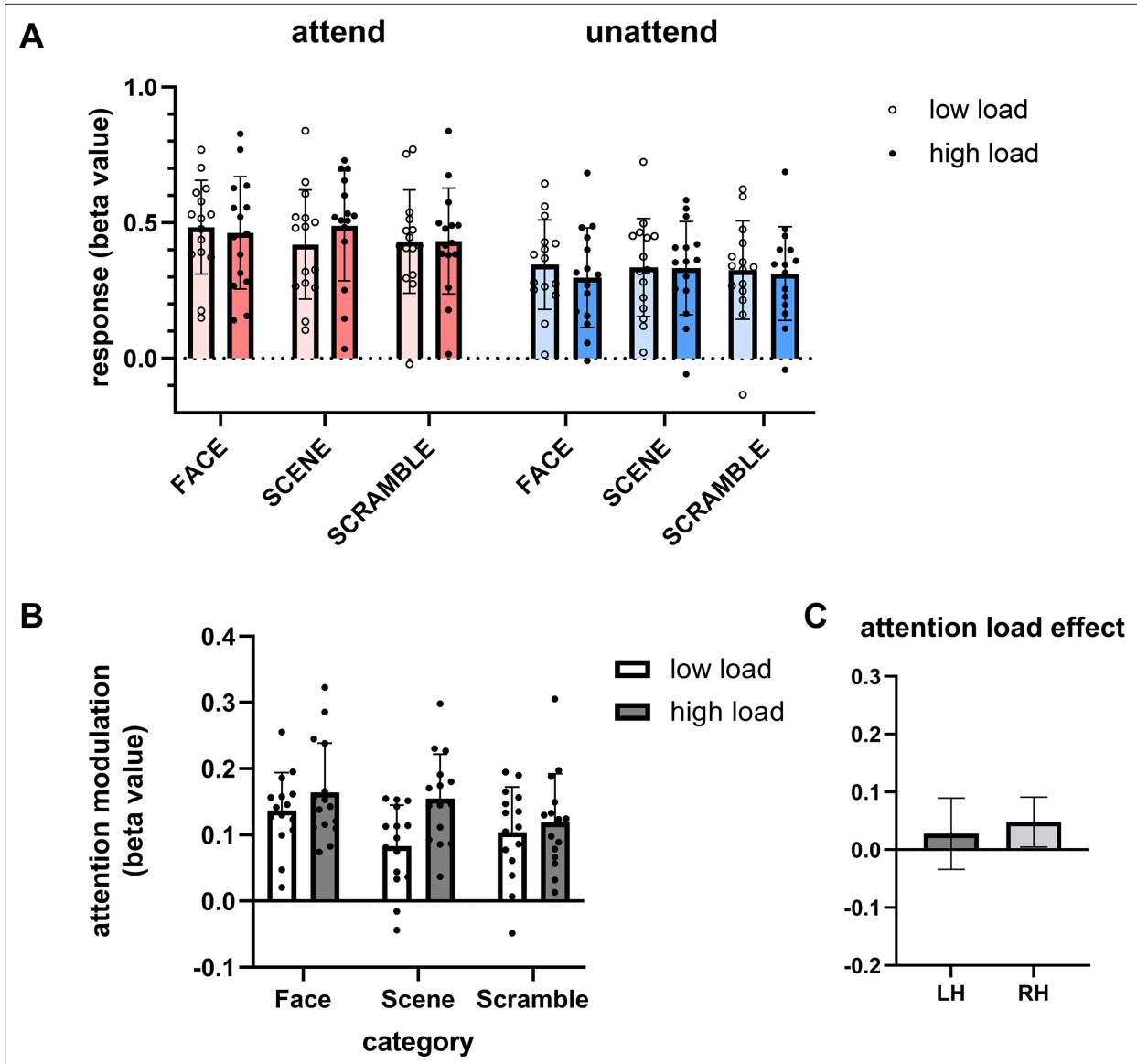

**Figure 6.** Response and attentional modulation of human posterior inferotemporal cortex (hPIT) to images of different categories. (**A**) The activation level of hPIT when attending to or not attending to images of different categories. Red bars indicate the condition with attention in the receptive field (attended), while blue bars indicate the condition with attention on the other side (unattended). Bars with higher chroma represent high load condition, lower chroma represent low load condition. (**B**) The modulation pattern of hPIT when presenting different categories of images with different attentional load. White bars represent the condition with low-load attention. Gray bars represent condition with high-load attention. (**C**) The attentional load effect (high load – low load) of bilateral hPIT, averaged across three categories. Error bars indicate 95% confidence interval (n=15).

image categories. The observation that the attentional modulation in hPIT is sensitive to task load further supports hPIT's role as an attentional priority map.

## Functional connectivity of hPIT to attentional networks

To unravel the functional connectivity of hPIT with the whole brain (especially the nodes of attentional network), resting-state data were analyzed, with ROIs defined by task-based data (Experiments 1 and 3, *Figure 7A*) and cortical parcellation atlas (*Glasser et al., 2016*). First, to visualize the patterns of functional connectivity, we calculated the average time course of hPIT, FFA, and LOp using small sphere-shaped ROIs (2 mm radius) to avoid signal contamination for better visualization and obtained the correlation coefficient between the seeds and all other voxels of brain to generate functional connectivity maps from the three spatially restricted seeds (*Figure 7B*). It is apparent that

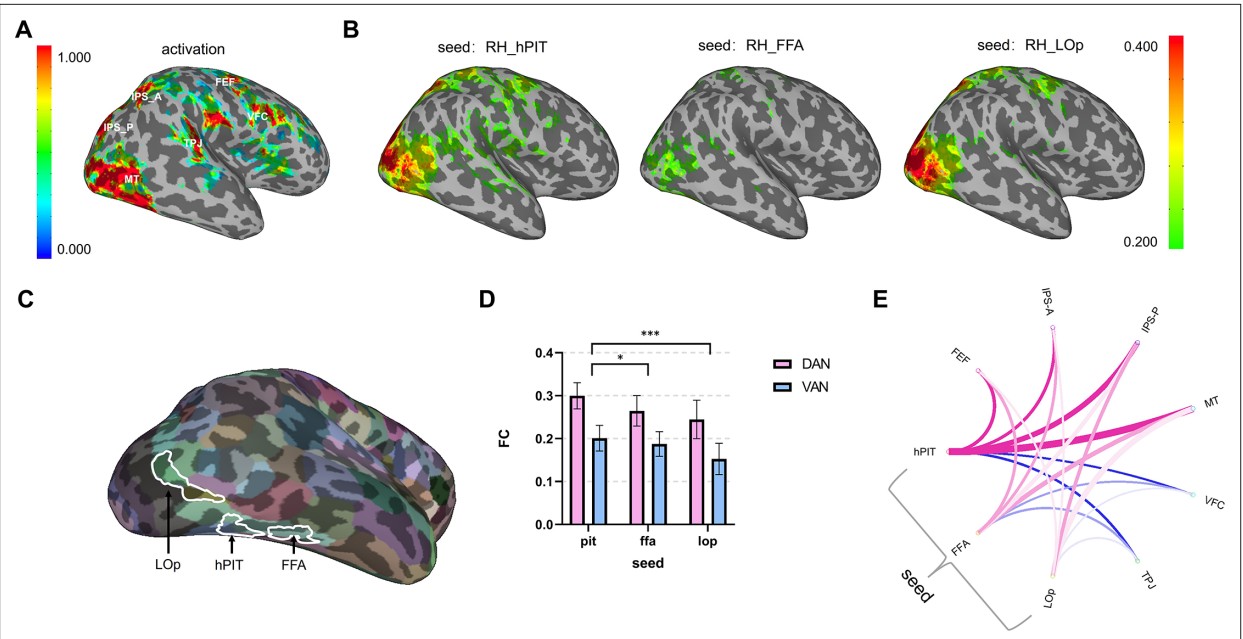

**Figure 7.** Functional connectivity analysis of human posterior inferotemporal cortex (hPIT) and its neighboring areas. (**A**) Activation map showing beta of contrast (attend contra moving dot – baseline) (p<0.01, uncorrected) and the location of critical nodes in dorsal and ventral attentional network (VAN) on one typical subject (S03). (**B**) Thresholded map showing functional connectivity of seed sphere right-hemi hPIT, right-hemi fusiform face area (FFA), and right-hemi lateral-occipital cortex (LOp) (Spearman's rank correlation coefficient >0.2, p<0.05, uncorrected), averaged across subjects and projected onto the surface of standard brain MNI152_2009c. Color bar attached indicates the intensity of activation (**A**) and correlation (**B**). (**C**) The relative location of LOp, hPIT, and FFA on the inflated cortical surface of parcellation map (Glasser's atlas). (**D**) Strength of functional connectivity of seed hPIT, FFA, and LOp with DAN and VAN. Error bars indicate 95% confidence interval (n=15). *** indicates the significance of p<0.001. * indicates the significance of p<0.05. (**E**) Circular plot for functional connectivity of seed hPIT, FFA, and LOp with nodes of attentional network of right hemisphere, with pink lines indicating connection with nodes of DAN, blue lines indicating nodes of VAN. Connections to left hemisphere nodes show similar but weaker trends. Opacity of each line connecting seed and nodes represents the rank of its connectivity strength (the strongest 100%, the middle 44%, the weakest 11%). The width of each line is scaled to its cubic numerical intensities of connectivity.

hPIT had strong functional connections with nodes of DAN (FEF, IPS, MT) and VAN (VFC, TPJ). Then, the functional correlation using whole ROIs (hPIT, FFA, and LOp, example shown in *Figure 7C*) with bilateral attentional network nodes was calculated on each participant. The connection strength of bilateral hPIT, FFA, and LOp with DAN and VAN was compared using two-way ANOVA (n=15, attentional networks × seed ROIs), and results showed significant main effects of seed ROIs (F(1.732, 24.25)=8.653, p=0.0021) and attentional networks (F(1.000, 14.00)=60.56, p<0.0001) (*Figure 7D*). Specifically, the correlation coefficients of hPIT with attentional networks were stronger than the other two seed areas (Dunnett, one-sided): PIT vs. FFA (q(14)=2.135, adjusted p=0.0452), PIT vs. LOp (q(14)=4.725, adjusted p=0.0003). As for hemispheric differences, a two-way ANOVA (hemisphere [left hPIT, right hPIT]×attentional network [left DAN, right DAN, left VAN, right VAN]) revealed a significant main effect of attentional network [F(2.120, 29.67)=37.38, p<0.0001], but no significant main effect of hPIT hemisphere was observed (F(1.000, 14.00)=3.129, p=0.0987) or the interaction (F(1.266, 17.72)=0.3339, p=0.6223).

We also performed psychophysiological interaction using data from Experiment 2, which revealed task-modulated right hPIT connectivity. During attended dot conditions (contrast [attend contralateral dot – attend ipsilateral dot]), right hPIT showed significantly enhanced functional coupling with FFA (t (14)=2.15, p=0.048) and LOp (t(14)=2.50, p=0.029), with trending enhancement to TPJ (t(14)=1.85, p=0.088). This suggests hPIT may allocate attentional resources to object-processing regions following priority map generation.

The stronger connection of hPIT with each attentional node could be visualized in a circular connectivity plot (*Figure 7E*). The strong functional connections of hPIT with attentional networks, especially the DAN, support the important role it plays in attentional control.

## Discussion

It is traditionally accepted that the networks of frontal and parietal cortex play important roles in the control and engagement of endogenous and exogenous attention, while regions of occipito-temporal cortex specialize in the processing of colors and shapes, leading to the representation of scenes and objects, such as faces, bodies, and words (*Kastner and Ungerleider, 2000*; *McCandliss et al., 2003*; *Moore et al., 2003*; *Peelen and Downing, 2005*; *Lafer-Sousa and Conway, 2013*; *Baldauf and Desimone, 2014*; *Grill-Spector and Weiner, 2014*; *Moore and Zirnsak, 2017*; *Stemmann and Freiwald, 2019*). Our results provide new distinct insights about the role of the hPIT in the control and implementation of attention. We identified a specific area within the PITd, hPIT, where its activation exhibited little category selectivity, but was strongly modulated by attention across tasks, and reflected a combined effect of top-down attention and bottom-up attention. Further, the attentional modulation in hPIT was significantly affected by attentional load, as well as image category, consistent with the nature of attention. In addition, functional connectivity analysis revealed that hPIT, compared to its neighbor areas FFA and LOp, exhibited stronger connectivity with nodes of the attentional network. Taken together, these results suggest that hPIT fulfills the key criteria of an attentional priority map. It shows spatially specific responses that are strongly modulated by attention, relatively invariant to stimulus features, and functionally integrated with both DAN and VAN. Although classical regions such as LIP and FEF are often associated with attentional control, they are less consistent with our more stringent priority map criteria in the present study, as their modulation was mainly driven by top-down attention. Regarding input integration, hPIT is situated closer to high-level visual areas, making it well positioned to integrate bottom-up saliency information with top-down signals.

The hPIT is in close proximity to other object-processing regions in the inferotemporal cortex, including the FFA, the visual word form area (VWFA), and lateral occipital complex (LOC). The hPIT ((−34, −72, −14), (33, −73, −13)) is posterior to the FFA (40, −55, −10) (*Kanwisher et al., 1997*) and VWFA (−43, −56, −16) (*Rauschecker et al., 2012*; *Zhang et al., 2018*). LOC is localized by more activation when viewing objects than scramble, consisting of two subregions: LO (in our paper, LOp) and Loa/pFs (*Grill-Spector et al., 2001*). The hPIT is inferior to the LOp (bound by (−41, −77, 3) and (−36, −71, −13)), posterior to the Loa/pFs (−38,−50, −17) (*Grill-Spector et al., 1999*).

Why does a brain region in the inferotemporal cortex exhibit the properties of an attentional priority map, integrating the top-down and bottom-up attention? Previous studies have suggested that LIP/IPS, FEF, and SC are closely linked to attentional guidance and eye movement and controls: LIP as a representation of attentional priority that remaps across saccades, FEF as an eye movement controller receiving LIP responses, and SC as reflecting the final saccade (*Andersen et al., 1992*; *Thompson et al., 1996*; *McPeek and Keller, 2002*; *Shen and Paré, 2007*; *Buschman and Miller, 2009*; *Arcizet et al., 2011*; *Zhou and Desimone, 2011*; *Foley et al., 2017*; *Bisley and Mirpour, 2019*). On the other hand, both top-down and bottom-up attention are frequently associated with objects; ideally, an area serving as attentional priority map should be positioned where it is easy to assess object information and also has broad connections with other regions of the attentional networks. The hPIT's location in the inferotemporal cortex facilitates the integration of key information from adjacent areas specialized in object processing. As shown in our functional connectivity results, as well as in previous studies (*Bogadhi et al., 2019*; *Sani et al., 2019*; *Sani et al., 2021*; *Boshra and Kastner, 2022*), hPIT is connected with key areas of attentional and eye movement controls, such as LIP/IPS, FEF, and SC. The placement of an area serving attentional priority map in the inferotemporal cortex, i.e., hPIT, has strategic advantages for both bottom-up information transmission and top-down attentional modulation.

In addition to the strong connection between hPIT and the DAN, consistent with a recent study (*Meng et al., 2024*), our functional connectivity analysis also revealed positive connections with TPJ and VFC, as shown in individual connectivity maps, though these connections were weaker compared to those with dorsal attentional nodes. The connectivity with both the DAN and VAN further supports hPIT's unique role in bridging the bottom-up and top-down attention.

Our study is limited in that the effects of feature-based attention were not specifically examined in our experiments. Besides, although faces used in the task had neutral expressions and the scene pictures were also neutral, we cannot exclusively eliminate the possibility that potential semantic familiarity or emotional salience may contribute to the subtle category-related effects in the results of Experiment 3. In addition, our current study lacks the temporal dynamic information of processing

in hPIT and its downstream and upstream brain areas due to the use of fMRI measurements. Future studies would benefit from utilizing imaging methods with higher time resolution, with designs that address both spatial and feature-based attention.

In summary, the hPIT showed strong attentional modulation across stimuli and tasks, with sensitivity to attentional load and more robust attentional modulation in the presence compared to the absence of visual input. The hPIT also demonstrated functional connectivity to both DAN and VAN. Together, our findings demonstrate the distinct role of hPIT in attentional control, namely as an attentional priority map that integrates endogenous and exogenous attentional processes.

## Acknowledgements

This work was supported by STI2030-Major Projects (Grant Nos. 2021ZD0204200 and 2021ZD0203800) and Key Research Program of Frontier Sciences, Chinese Academy of Science (Grant No. KJZD-SW-L08).

## Additional information

### Funding

| Funder | Grant reference number | Author |
| --- | --- | --- |
| Ministry of Science and Technology of the People's Republic of China | 2021ZD0204200 | Sheng He |
| Ministry of Science and Technology of the People's Republic of China | 2021ZD0203800 | Sheng He |
| Key Research Program of Frontier Sciences, Chinese Academy of Science | KJZD-SW-L08 | Sheng He |

The funders had no role in study design, data collection and interpretation, or the decision to submit the work for publication.

### Author contributions

Siyuan Huang, Conceptualization, Data curation, Software, Formal analysis, Validation, Investigation, Visualization, Methodology, Writing – original draft, Project administration, Writing – review and editing; Lan Wang, Conceptualization, Data curation, Supervision, Methodology, Project administration, Writing – review and editing; Sheng He, Conceptualization, Resources, Data curation, Supervision, Funding acquisition, Methodology, Project administration, Writing – review and editing

### Author ORCIDs

Siyuan Huang https://orcid.org/0009-0004-2347-293X
Lan Wang https://orcid.org/0000-0002-8140-1011
Sheng He https://orcid.org/0000-0001-5547-923X

### Ethics

All participants provided written informed consent prior to participation. The experimental protocol was approved by the Institutional Review Board of the Institute of Biophysics, Chinese Academy of Sciences (approval number: 2020-IRBH-001), and all procedures were conducted in accordance with the Declaration of Helsinki.

Reviewer #1 (Public review): https://doi.org/10.7554/eLife.107111.3.sa1
Reviewer #2 (Public review): https://doi.org/10.7554/eLife.107111.3.sa2
Author response https://doi.org/10.7554/eLife.107111.3.sa3

# Additional files

### Supplementary files

Supplementary file 1. Exact location and cluster size of human posterior inferotemporal cortex (hPIT) in every subject's cortical surface.

Supplementary file 2. Summary table of attentional modulation (signal change%) across ROIs (human posterior inferotemporal cortex [hPIT], V1, medial temporal visual area [MT], intraparietal sulcus [IPS], frontal eye fields [FEF], temporal parietal junction [TPJ], ventral frontal cortex [VFC], fusiform face area [FFA], lateral-occipital cortex [LOp]) under both blank and dot conditions.

Supplementary file 3. Results of one-sample t-test measuring the modulation of attention by beta value of contrast (attend contralateral – attend ipsilateral).

MDAR checklist

### Data availability

fMRI data supporting the findings of this study have been deposited in figshare and are publicly available at https://doi.org/10.6084/m9.figshare.27230934.

The following dataset was generated:

| Author(s) | Year | Dataset title | Dataset URL | Database and Identifier |
|---|---|---|---|---|
| Huang S | 2024 | MRI processed data for 15 subjects * 3 sessions | https://doi.org/10.6084/m9.figshare.27230934 | figshare, 10.6084/m9.figshare.27230934 |

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
