## [Editor Report · eLife Assessment]

This **important** study reports that the human posterior inferotemporal cortex (hPIT) functions as an attentional priority map, integrating both top-down and bottom-up attentional signals rather than serving solely as an object-processing region. The experiments and analyses are well conducted and provide **compelling** evidence that hPIT bridges dorsal and ventral attention networks and is robustly modulated by attention across diverse visual tasks. The study will be relevant for researchers investigating visual attention, high-level visual cortex, and the neural mechanisms that integrate endogenous and exogenous attentional control.

---

## [Referee Report · Reviewer #1 (Public review)]

The manuscript titled "The distinct role of human PIT in attention control" by Huang et al. investigates the role of the human posterior inferotemporal cortex (hPIT) in spatial attention. Using fMRI experiments and resting-state connectivity analyses, the authors present compelling evidence that hPIT is not merely an object-processing area, but also functions as an attentional priority map, integrating both top-down and bottom-up attentional processes. This challenges the traditional view that attentional control is localized primarily in frontoparietal networks.

The manuscript is strong and of high potential interest to the cognitive neuroscience community. Below, I raise questions and suggestions to help with the reliability, methodology, and interpretation of the findings.

(1) The authors argue that hPIT satisfies the criteria for a priority map, but a clearer justification would strengthen this claim. For example, how does hPIT meet all four widely recognized criteria, such as spatial selectivity, attentional modulation, feature invariance, and input integration, when compared to classical regions such as LIP or FEF? A more systematic summary of how hPIT meets these benchmarks would be helpful. Additionally, to what extent are the observed attentional modulations in hPIT independent of general task difficulty or behavioral performance?

(2) The authors report that hPIT modulation is invariant to stimulus category, but there appear to be subtle category-related effects in the data. Were the face, scene, and scrambled images matched not only in terms of luminance and spatial frequency, but also in terms of factors such as semantic familiarity and emotional salience? This may influence attentional engagement and bias interpretation.

(3) The result that attentional load modulates hPIT is important and adds depth to the main conclusions. However, some clarifications would help with the interpretation. For example, were there observable individual differences in the strength of attentional modulation? How consistent were these effects across participants?

(4) The resting-state data reveal strong connections between hPIT and both dorsal and ventral attention networks. However, the analysis is correlational. Are there any complementary insights from task-based functional connectivity or latency analyses that support a directional flow of information involving hPIT? In addition, do the authors interpret hPIT primarily as a convergence hub receiving input from both DAN and VAN, or as a potential control node capable of influencing activity in these networks? Also, were there any notable differences between hemispheres in either the connectivity patterns or attentional modulation?

(5) A few additional questions arise regarding the anatomical characteristics of hPIT: How consistent were its location and size across participants? Were there any cases where hPIT could not be reliably defined? Given the proximity of hPIT to FFA and LOp, how was overlap avoided in ROI definition? Were the functional boundaries confirmed using independent contrasts?

Comments on revisions:

The authors have successfully addressed my previous questions and concerns. The public comments above reflect my views on the initial submission and, in my opinion, will remain helpful for general readers. Given this, I do not have additional public comments and will keep my previous public review unchanged.

---

## [Referee Report · Reviewer #2 (Public review)]

Summary

This study investigates the role of the human posterior inferotemporal cortex (hPIT) in attentional control, proposing that hPIT serves as an attentional priority map that integrates both top-down (endogenous) and bottom-up (exogenous) attentional processes. The authors conducted three types of fMRI experiments and collected resting-state data from 15 participants. In Experiment 1, using three different spatial attention tasks, they identified the hPIT region and demonstrated that this area is modulated by attention across tasks. In Experiment 2, by manipulating the presence or absence of visual stimuli, they showed that hPIT exhibits strong attentional modulation in both conditions, suggesting its involvement in both bottom-up and top-down attention. Experiment 3 examined the sensitivity of hPIT to stimulus features and attentional load, revealing that hPIT is insensitive to stimulus category but responsive to task load - further supporting its role as an attentional priority map. Finally, resting-state functional connectivity analyses showed that hPIT is connected to both dorsal and ventral attention networks, suggesting its potential role as a bridge between the two systems. These findings extend prior work on monkey PITd and provide new insights into the integration of endogenous and exogenous attention.

Strength

(1) The study is innovative in its use of specially designed spatial attention tasks to localize and validate hPIT, and in exploring the region's role in integrating both endogenous and exogenous attention, as prior works focus primarily on its involvement in endogenous attention.

(2) The authors provided very comprehensive experiment designs with clear figures and detailed descriptions.

(3) A broad range of analyses was conducted to support the hypothesis that hPIT functions as an attentional priority map -- including experiments of attentional modulation under both top-down and bottom-up conditions, sensitivity to stimulus features and task load, and resting-state functional connectivity. These analyses showed consistent results.

(4) Multiple appropriate statistical analyses - including t-tests, ANOVAs, and post-hoc tests-were conducted, and the results are clearly reported.

Comments on revisions:

The authors have addressed our comments in their revised manuscript and in their response to the reviewers. We don't have any further suggestions or comments.

---

## [Author Response]

The following is the authors’ response to the original reviews.

**Public Reviews:**

**Reviewer #1 (Public review):**
The manuscript titled "The distinct role of human PIT in attention control" by Huang et al. investigates the role of the human posterior inferotemporal cortex (hPIT) in spatial attention. Using fMRI experiments and resting-state connectivity analyses, the authors present compelling evidence that hPIT is not merely an object-processing area, but also functions as an attentional priority map, integrating both top-down and bottom-up attentional processes. This challenges the traditional view that attentional control is localized primarily in frontoparietal networks.The manuscript is strong and of high potential interest to the cognitive neuroscience community. Below, I raise questions and suggestions to help with the reliability, methodology, and interpretation of the findings.

Thank you for a nice summary of the key points of our study. Below you will find our reply to your questions.

(1) The authors argue that hPIT satisfies the criteria for a priority map, but a clearer justification would strengthen this claim. For example, how does hPIT meet all four widely recognized criteria, such as spatial selectivity, attentional modulation, feature invariance, and input integration, when compared to classical regions such as LIP or FEF? A more systematic summary of how hPIT meets these benchmarks would be helpful. Additionally, to what extent are the observed attentional modulations in hPIT independent of general task difficulty or behavioral performance?

Great suggestions! For the first suggestion, we have included a clearer justification in the discussion part of manuscript (line 405-406). For the second one, all participants received task practice prior to scanning, and task accuracy exceeded 90%, suggesting the tasks were not overly demanding. Although ceiling effects limit the interpretability of behavioral-performance correlations, we argue that higher task demands would likely require greater attentional effort, leading to stronger modulation in hPIT, which aligns with our findings.

(2) The authors report that hPIT modulation is invariant to stimulus category, but there appear to be subtle category-related effects in the data. Were the face, scene, and scrambled images matched not only in terms of luminance and spatial frequency, but also in terms of factors such as semantic familiarity and emotional salience? This may influence attentional engagement and bias interpretation.

The response of hPIT is not sensitive to stimulus category, but attentional modulation in hPIT is slightly stronger to faces than scenes and scrambled images. Although faces used in the task had neutral expressions and the scene pictures were also neutral, we acknowledge that we indeed cannot exclusively eliminate the possibility that potential semantic familiarity or emotional salience may contribute to the subtle category-related effects in the results of experiment 3. This limitation has been noted in the discussion part of manuscript (line 440-442).

(3) The result that attentional load modulates hPIT is important and adds depth to the main conclusions. However, some clarifications would help with the interpretation. For example, were there observable individual differences in the strength of attentional modulation? How consistent were these effects across participants?

Yes, individual differences exist. In the manuscript, we have included individual subject data points in the figure 6B. No data exceeded three standard deviations from the group mean, suggesting that the attentional modulation effects were generally consistent across participants.

(4) The resting-state data reveal strong connections between hPIT and both dorsal and ventral attention networks. However, the analysis is correlational. Are there any complementary insights from task-based functional connectivity or latency analyses that support a directional flow of information involving hPIT? In addition, do the authors interpret hPIT primarily as a convergence hub receiving input from both DAN and VAN, or as a potential control node capable of influencing activity in these networks? Also, were there any notable differences between hemispheres in either the connectivity patterns or attentional modulation?

Though it’s hard to generate directional flow of information from fMRI due to the low temporal resolution. We agree that besides resting-state connection, task-based functional connectivity analyses would have the potential to provide additional information about whether hPIT serves as a convergence node or a control hub. We have conducted task-based functional connectivity analyses, specifically PPI, using data from experiment 2, which revealed task-modulated right hPIT connectivity with FFA, LOp, and TPJ, suggesting hPIT may allocate attentional resources to object-processing regions following priority map generation (line 378-383). Given the limited number of significant PPI results and the inherent constraints of fMRI in capturing fast or transient attention-related interactions, the present data do not allow us to determine the role of hPIT. Future studies combining effective connectivity or causal perturbation methods (e.g., DCM, TMS-fMRI) would be ideal to test whether hPIT acts as a control node influencing activity within DAN and VAN.

We also observed modest hemispheric asymmetries in connectivity—for instance, both left and right hPIT showed stronger connectivity with right-hemisphere attention nodes. This has been described in the results part of manuscript (line 373-377).

(5) A few additional questions arise regarding the anatomical characteristics of hPIT: How consistent were its location and size across participants? Were there any cases where hPIT could not be reliably defined? Given the proximity of hPIT to FFA and LOp, how was overlap avoided in ROI definition? Were the functional boundaries confirmed using independent contrasts?

We can see a relatively consistent size and location of hPIT across subjects in Supplementary Figure 1, where the voxel size and location for individual subjects reported. The consistency also demonstrated by figure 4C.

We avoided overlap with the FFA and LOp by manually delineating the hPIT which is defined by conjunction maps across three tasks and by avoiding overlapping voxels. The FFA was defined using an independent contrast (Exp3 contrast [face-scene]) and the Lop location was defined by anatomical parcellation (Glasser et al., 2016).

**Reviewer #2 (Public review):**
SummaryThis study investigates the role of the human posterior inferotemporal cortex (hPIT) in attentional control, proposing that hPIT serves as an attentional priority map that integrates both top-down (endogenous) and bottom-up (exogenous) attentional processes. The authors conducted three types of fMRI experiments and collected resting-state data from 15 participants. In Experiment 1, using three different spatial attention tasks, they identified the hPIT region and demonstrated that this area is modulated by attention across tasks. In Experiment 2, by manipulating the presence or absence of visual stimuli, they showed that hPIT exhibits strong attentional modulation in both conditions, suggesting its involvement in both bottom-up and top-down attention. Experiment 3 examined the sensitivity of hPIT to stimulus features and attentional load, revealing that hPIT is insensitive to stimulus category but responsive to task load - further supporting its role as an attentional priority map. Finally, resting-state functional connectivity analyses showed that hPIT is connected to both dorsal and ventral attention networks, suggesting its potential role as a bridge between the two systems. These findings extend prior work on monkey PITd and provide new insights into the integration of endogenous and exogenous attention.Strengths(1) The study is innovative in its use of specially designed spatial attention tasks to localize and validate hPIT, and in exploring the region's role in integrating both endogenous and exogenous attention, as prior works focus primarily on its involvement in endogenous attention.(2) The authors provided very comprehensive experiment designs with clear figures and detailed descriptions.(3) A broad range of analyses was conducted to support the hypothesis that hPIT functions as an attentional priority map -- including experiments of attentional modulation under both top-down and bottom-up conditions, sensitivity to stimulus features and task load, and resting-state functional connectivity. These analyses showed consistent results.(4) Multiple appropriate statistical analyses - including t-tests, ANOVAs, and post-hoc tests - were conducted, and the results are clearly reported.

Thank you for a nice summary of the key points and strengths of our study.

Weaknesses(1) The sample size is relatively small (n = 15), and inter-subject variability is big in Figures 5 and 6, as seen in the spread of individual data points and error bars. The analysis of attention-modulated voxel map intersections appears to be influenced by multiple outliers.

We agree that the sample size (n = 15) is not ideal, and we acknowledge that some data points in Figures 5 and 6 appear to be potential outliers. However, according to conventional outlier detection criteria, all data points fell within three standard deviations of the group mean and were therefore retained for analysis.

Moreover, the attention-modulated voxel intersection map shown in Figure 4C is insensitive to outliers, because the intersection plotted is based on the number of subjects

(2) The authors acknowledge important limitations, including the lack of exploration of feature-based attention and the temporal constraints inherent to fMRI.

Yes, we have mentioned these limitations in the discussion.

(3) Prior research has established that regions such as the prefrontal cortex (PFC) and posterior parietal cortex (PPC) are involved in both endogenous and exogenous attention and have been proposed as attentional priority maps. It remains unclear what is uniquely contributed by hPIT, how it functionally interacts with these classical attentional hubs, and whether its role is complementary or redundant. The study would benefit from more direct comparisons with these regions.

In this study, we define the ROI base on intersection across three different types of spatial attention tasks, which is a stricter criterion. And the results didn’t reveal spatial attentional modulation across tasks besides PITd. This could be due to the lack of lateralized responses in PFC/PPC. To evaluate whether a region qualifies as a priority map, we applied four widely accepted criteria (as mentioned in introduction). While dorsal and ventral attention network (DAN and VAN) regions can be considered supportive components of the priority map system, our findings suggest that among the regions tested, only hPIT fully meets all criteria. In Experiment 2, we included regions such as VFC (as part of PFC) and IPS (as part of PPC), and our findings suggest these areas are more involved in top-down attention. In the revision, we have performed additional analysis on PPC (IPS) and PFC (FEF, VFC), shown in Figure S2.

(4) The functional connectivity analysis is only performed on resting-state data, and this approach does not capture context-dependent interactions. Task-based data analysis can provide stronger evidence.

We acknowledge that resting-state FC is limited in assessing task-specific communication. To further investigate the role of hPIT, we have conducted PPI analysis, which revealed task-modulated right hPIT connectivity in attention allocation (line 378-383).

(5) The study does not report whether attentional modulation in hPIT is consistent across the two hemispheres. A comparison of hemispheric effects could provide important insight into lateralization and inter-individual variability, especially given the bilateral localization of hPIT.

We thank the reviewer for this suggestion. hPIT was localized bilaterally using the same intersection-based method in Experiment 1. We have now performed additional analysis and found hemispheric differences in hPIT attentional modulation (Experiment 2). Besides, we also found in Experiment 3, the difference of load modulation (averaged across stimulus categories) in left and right hPIT was not significant. These results have been reported in the results part of manuscript (line 347-351).